# Highly Specialized Mechanisms for Mitochondrial Transport in Neurons: From Intracellular Mobility to Intercellular Transfer of Mitochondria

**DOI:** 10.3390/biom13060938

**Published:** 2023-06-03

**Authors:** Marta Zaninello, Camilla Bean

**Affiliations:** 1Institute for Genetics, University of Cologne, 50931 Cologne, Germany; 2Cologne Excellence Cluster on Cellular Stress Responses in Aging-Associated Diseases (CECAD), 50931 Cologne, Germany; 3Department of Medicine, University of Udine, 33100 Udine, Italy

**Keywords:** cytoskeleton, microtubules, mitochondria, neuron, transport, TNTs, mitochondrial transplantation

## Abstract

The highly specialized structure and function of neurons depend on a sophisticated organization of the cytoskeleton, which supports a similarly sophisticated system to traffic organelles and cargo vesicles. Mitochondria sustain crucial functions by providing energy and buffering calcium where it is needed. Accordingly, the distribution of mitochondria is not even in neurons and is regulated by a dynamic balance between active transport and stable docking events. This system is finely tuned to respond to changes in environmental conditions and neuronal activity. In this review, we summarize the mechanisms by which mitochondria are selectively transported in different compartments, taking into account the structure of the cytoskeleton, the molecular motors and the metabolism of neurons. Remarkably, the motor proteins driving the mitochondrial transport in axons have been shown to also mediate their transfer between cells. This so-named intercellular transport of mitochondria is opening new exciting perspectives in the treatment of multiple diseases.

## 1. Introduction

Neurons are polarized structures divided into compartments, which are functionally distinct units. Organelles, proteins and RNA are transported along neuronal processes—named axons and dendrites—to distal areas such as synapses, growth cones and branching points where mitochondria play different roles. Mitochondria are abundant in presynapses, while roughly 10% of dendritic spines contain mitochondria in basal conditions [1]. Neuronal activity increases the transport of mitochondria in synapses [1]. At a wider level, the somatodendritic and the axonal compartments have different distribution and trafficking properties in terms of cargo vesicles and organelles, which do not diffuse but are actively selected at the pre-axonal exclusion zone [2]. The majority of mitochondria are stationary in axons across different species [3]. The other motile mitochondria travel long distances and the direction of the transport is named anterograde or retrograde if it is from the cell body to the distal area of axons or vice versa. Motile mitochondria show complex trajectories, including linear and oscillatory, with pauses and changes in direction [3]. The fraction of mitochondria in a moving or a stationary state is associated with axonal growth: in the region of active growth cones, there is a motile-to-stationary shift of mitochondria that is reversed when axonal growth is blocked. The consequence of this dynamic balance is that the net transport is anterograde in growing axons and retrograde in blocked axons [4]. In this review, we focus on the mechanisms regulating the complex transport of mitochondria along the extraordinary distances over which neurons can extend.

Another type of long-distance transport contributing to the maintenance of neural homeostasis is represented by the transfer of mitochondria between adjacent cells. Different processes drive this transcellular communication, including the formation of tunneling nanotubes (TNTs). These structures, described for the first time twenty years ago by Rustom and colleagues [5], seem to rely on the same molecular system used for mitochondrial transport and docking in axons. The intracellular transmission of mitochondria observed in astrocytes, microglia, and neurons, is important for the recovery of neural functions supporting both their viability and post-injury recovery [6,7,8,9]. Similarly to spontaneous mitochondrial transfer between cells, stem cell-derived mitochondrial transplantation can provide an exogenous mitochondrial source thus restoring the mitochondrial functions in recipient cells [10,11]. As the primary hallmark in a wide range of brain states and pathologies is mitochondrial dysfunction, mitochondrial delivery into injured cells is opening a novel horizon for treating many diseases. Here, we summarize the clinical applications highlighting the opportunities and challenges of mitochondrial transfer/transplantation, especially in brain disorders.

## 2. Mitochondria Move along Microtubules and Actin Filaments

The cytoskeleton in neurons is composed of microtubules, actin filaments and neurofilaments. Neurofilaments are enriched in axons where they determine the diameter and the conductance. Microtubules and actin filaments regulate axonal maturation and growth and build the support for the transport of organelles, including mitochondria [12] (Figure 1).

### 2.1. Microtubules

Microtubules are tubes assembled by dimers of α- and β-tubulin oriented head-to-tail in rapid phases of growth and collapse named “dynamic instability” [13]. These structures confer the distinct polarity of microtubules, where α- and β-tubulins are exposed at the minus- and plus-end of microtubules, respectively. Microtubules have a peculiar organization and dynamics in neurons: they present a plus-end toward the distal part of axons and have a mixed polarity in dendrites [14]. Microtubules polymerize at the plus-end via the incorporation of fresh guanosine triphosphate (GTP) to β-tubulin, which is hydrolyzed to guanosine diphosphate (GDP) in already incorporated tubulin dimers. However, GTP-bound tubulin dimers have also been described in the stable microtubule lattice [15,16,17] and are more enriched in axons than dendrites [17,18]. These so-named GTP islands protect microtubule depolymerization and promote self-repair [19,20] but also regulate the local conformation of tubulin to modulate the transport of mitochondria [18]. A recent study showed that anterograde mitochondria halt along GTP-bound elongated dimers within the microtubule bundle but they remain motile at the rim of the microtubule bundle [18]. Furthermore, the affinity of the motor proteins kinesins linking organelles to microtubules depends on the different conformation of GDP- or GTP-bound dimers [17,21,22]. For mitochondria, these elongated GTP islands increase the velocity of the Kinesin motor for mitochondria KIF5B [18].

Post-translational modifications of tubulin appear after the polymerization of microtubules and include acetylation, detyrosination, glycylation and glutamylation (Figure 1a). Developing axons grow in response to attractive and repulsive chemical guidance clues. This process is highly dynamic and involves cycles of the de- and re-polymerization of actin and microtubules in the terminal part of axons, named the growth cone. In an initial phase, actin extends in protrusions, which are later invaded by microtubules and organelles as mitochondria and endoplasmic reticulum (engorgement). No acetylation is detected in growth cones, consistent with the presence of highly dynamic microtubules [23]. Finally, the new formed structure is consolidated by the depolymerization of actin and stabilization of microtubules [24]. This process is regulated by kinases activated by growth factors such as Slit, Wnt or nerve growth factor (NGF) bound to the receptors roundabout (Robo), trafficking kinesin protein (Trak) A and frizzled (Frz)/low-density lipoprotein receptor-related protein (LRP), respectively. These glycogen synthase kinase 3β (GSK3β) and Abl kinases regulate the localization of plus-end tracking proteins such as cytoplasmic linker-associated protein 2 (Clasp2), adenomatous polyposis coli (APC) and microtubule-associated protein 1B (Map1B), which alter the stability of microtubules [25]. Differentiated axons exhibit stable microtubules with high acetylation, glutamylation and detyrosination [23]. In addition to their role in regulating the stability of microtubules, post-translational modifications of tubulin also regulate the affinity of motor adaptors to microtubules and alter the general transport of organelles rather than being specific to mitochondria [26,27]. Accessory proteins of microtubules regulate the transport of mitochondria in a similar way. For instance, Map1B knockout neurons increase the retrograde transport [28] and mitochondria of N2a cells, or primary neurons overexpressing Tau do not travel in the anterograde direction [29,30,31]. This is probably due to a combination of Tau that can generally destabilize microtubules [32] and inhibit the kinesin-dependent transport of vesicles and organelles [31]. Taken together, these studies seem to indicate that the transport of mitochondria is not specifically regulated at the level of microtubule organization. However, it is possible that the studies conducted so far analyzed the general movement of mitochondria (stationary over motile mitochondria) and that broader methods to dissect the complex motility of mitochondria could show a subtle but specific role of accessory proteins and post-translational modifications of tubulin [33].

### 2.2. Actin

Actin filaments are formed by polarized globular monomers bound via weak interactions. Thus, actin polymers are intrinsically unstable and difficult to visualize in neurons. Electron microscopy allows the visualization of patches of actin along axons, synapses and growth cones [12]. Very little is known about the role of actin in the long-range transport of mitochondria. It appears that microtubules do not have an exclusive role in this transport because the depolymerization of microtubules using nocodazole or vinblastine does not completely stop mitochondria in axons and dendrites [34,35]. In addition, the disruption of the actin cytoskeleton using cytochalasin-D or lantruculin B has no effect [34]. A more recent study shows that cytochalasin-D stabilizes axonal mitochondria and destabilizes dendritic mitochondria [36], suggesting that actin organization modulates the transport of mitochondria via different mechanisms in the two compartments. As we mentioned previously, actin is also present in synapses, where it mediates the docking of mitochondria after the administration of NGF [37], regulating the short-range transport of mitochondria.

## 3. Molecular Motors Transport Mitochondria via Microtubules

Mitochondria associate with the microtubule network through kinesin and dynein, the molecular motors that drive mitochondria in the anterograde and retrograde direction, respectively (Figure 1b,c). Therefore, kinesins move mitochondria towards the plus-ends of microtubules and dynein mediates minus-end-directed mitochondrial transport. Among the kinesin superfamily proteins (also known as KIFs) encoded in humans and mice by 45 different genes [38], KIF5/Kinesin-1, KIF1B/Kinesin-3 and Kinesin-Like Protein 6 (KLP6) are the main kinesins that mediate mitochondrial transport in neurons [39,40,41]. Two kinesin heavy chains (KHCs) and two kinesin light chains (KLCs) form a 380 kDa heterotetramer complex. This complex is composed of a conserved globular motor domain (or head) consisting of an adenosine triphosphate (ATP)-binding motif and a microtubule-binding domain, attached to a stalk domain for dimerization and to a tail domain with binding and regulatory functions. While the motor domains are highly conserved, the remaining sequences are unique for each kinesin, determining the cargo specificity and the direction of the transport. In contrast to the large specialized kinesin superfamily, only the cytoplasmatic Dynein 1 drives the minus-end-directed microtubule transport. Then, different mechanisms must explain how dynein specifically transports the numerous different cargoes. Dynein is a very large protein complex (1.2 MDa) composed of distinct polypeptides, all of which are present in two copies. The dynein heavy chain (DHC) contains the motor domain with six distinct AAA domains folded into a ring-shaped structure, together with a microtubule-binding domain and a tail for the assembly of the other components: the intermediate chains (DIC), the light intermediate chains (DLIC), and three different light chains (DLC) [42,43]. To be fully active, dynein requires interaction with the dynactin complex and a coiled-coil cargo adaptor [44]. Dynactin anchors dynein to microtubules through its larger subunit, p150Glued. In addition, dynactin contains two actin-related proteins, Arp1 and Actr10/Arp11, and the subunits p62, p25, and p27 [45]. The loss of Actr10 selectively reduces the mitochondrial retrograde transport, leading to the accumulation of mitochondria in axon terminals [45]. Specific adaptor proteins link kinesin and dynein–dynactin to each cargo for its transport along microtubules. On the outer mitochondrial membrane, the Mitochondrial Rho GTPase protein (Miro 1 and Miro 2; in humans, RHOT1 and RHOT2) recruits motor proteins by Milton/Trak adaptors and Metaxin (MTX) proteins. Miro contains a transmembrane domain in its C-terminus and two GTPase domains, one at the N-terminal and one near the C-terminal, flanking two calcium-binding EF hand motifs which regulate the motility of mitochondria depending on calcium. The GTPase domains are crucial for regulating mitochondrial distribution: when GDP is bound, Miro does not recruit adaptor and motor proteins [46]. In mammals, the different binding specificity of the two Trak proteins targets mitochondria to dendrites and axons; while Trak1 binds to both kinesin and dynein components and is responsible for the axonal localization of mitochondria, Trak2 primarily interacts with dynein and plays a critical role in targeting mitochondria to dendrites [47] (Figure 1b,c).

Interestingly, as in Miro1/2 double-knockout cells Traks are still recruited to the outer mitochondrial membrane to drive mitochondrial trafficking, Miro cannot be the only outer-membrane protein that links mitochondria to motor proteins [48]. In fact, the loss of Drosophila Miro (dMiro) cannot fully block mitochondrial movement [49]. Thus, several other kinesin-containing complexes have been discovered in neurons, including syntabulin (SYBU), fasciculation and elongation protein zeta 1 (FEZ1), or Ran-binding protein 2 (RanBP2) [50,51,52]. Interestingly all of these adaptors contribute to the maintenance and remodeling of synapses [53,54]. Another interactor of the Miro/Milton complex is Mitofusin 2 (Mfn2), the activity of which is disrupted in the presence of pathogenic mutants, which arrest mitochondria independently from Mfn2 profusion activity [55]. In contrast to the anterograde transport machineries, the players that link dynein to mitochondria are not well characterized. One suggested system is represented by the direct interaction of the outer mitochondrial membrane (OMM) protein voltage-dependent anion-selective channel (VDAC) with the dynein motor protein [56]. In conclusion, the main molecular actors involved in anterograde mitochondrial transport form a complex with Miro (receptor), KIF5 (motor) and Milton/Traks (adaptors). In contrast, dynein in complex with dynactin mediates retrograde mitochondrial transport by interacting with Milton/Trak2 and Miro.

## 4. Mitochondrial Docking and Anchoring Machineries in Neurons

Mitochondria localize in the specific regions of the neuron that need the most energy and a high ion flux, such as the synapses or the distal regions of actively growing axons [57]. While a large proportion of mitochondria move during neuronal development, in mature neurons the stationary pool of mitochondria represents more than two-thirds. Remarkably, more than 30% of the synapse is occupied by anchored mitochondria serving as ‘power stations’ [58]. In addition, as mitochondria can temporarily stop and start moving again at various subcellular localizations, specific docking mechanisms are needed. The mitochondria-associated protein Syntaphilin (SNPH) is a central microtubule calcium-dependent docking system (Figure 1d). SNPH directly interacts with microtubules through its N-terminal microtubule-binding domain while its C-terminal tail inserts into the OMM [59]. The dynein light chain LC8 binds to SNPH, thus enhancing the SNPH-based docking of mitochondria to microtubules [60]. SNPH can also interact with myosin VI (Myo6, Jaguar in *Drosophila*) and anchor mitochondria on presynaptic filamentous (F)-actin. The AMP-activated protein kinase (AMPK)-dependent phosphorylation of Myo6 drives the capture of mobile axonal mitochondria at presynaptic terminals, switching them from microtubule-dependent transport to actin-mediated tethering [61]. Other myosin motor proteins in association with the actin cytoskeleton have been shown to interrupt mitochondrial transport on the microtubule tracks, facilitating their positioning. For example, Myo19 stops mitochondria in actin filaments through Miro [62] (Figure 1d). In general, where mitochondria are more urgently needed, actin is enriched and docks mitochondria away from microtubules.

## 5. Metabolic Control of Mitochondrial Transport

Mitochondria produce ATP and buffer calcium for the functioning and survival of neurons. Thus, it is not surprising that mitochondria are more abundant in neurons than in other cell types and that calcium and adenosine diphosphate (ADP)/ATP are prime signaling molecules to regulate the distribution of mitochondria. Glutamate increases the amount of calcium and immobilizes mitochondria in synapses. A similar effect is obtained after injecting ADP or in hypoxia [63]. It seems that the localization of mitochondria is largely regulated by stationarity rather than transport, as the studies of signaling molecules have pointed out so far.

### 5.1. Calcium

As mentioned above, mitochondria rely on the adaptor proteins Milton in *Drosophila* and Traks in mammals for anterograde movement along axonal microtubules. These adaptors bridge the receptor proteins Miro1 and 2 located in the outer mitochondrial membrane to Kinesin-1 [62]. Miro contains two EF hands that bind calcium. In the absence of these domains, mitochondria fail to stop in the presence of calcium in axons and dendrites [62,64], placing Miro as a regulator of mitochondrial arrest in sites where cytoplasmic calcium concentration is high. Two models explain a conformational change in Miro in the presence of calcium, but the mechanism is the opposite (Figure 2a). According to Wang and Schwarz, Kinesin-1 binds microtubules with its N-terminal domain and the Miro/Milton complex via its C-terminal region [64]. This complex transports mitochondria, but calcium induces the sequestration of the whole complex from microtubules. In MacAskill et al., Miro directly binds Kinesin-1 and calcium detaches Kinesin-1 from mitochondria but not from microtubules [62]. In contrast, another study shows that Miro1 binds the mitochondrial calcium uniporter (MCU) through its N-terminal domain on the outer mitochondrial membrane [65] (Figure 2b). This interaction is required for the transport of mitochondria and depends on the MCU-dependent calcium influx in the mitochondrial matrix rather than cytoplasmic calcium [66]. E208K/E328K mutations in the EF hands of Miro abolish calcium entry in the matrix [66]. Curiously, a different mutation (R272Q) in the EF hands of Miro has no effect on mitochondrial movement, although it disrupts calcium handling in the mitochondria [67]. The discrepancies in the role of Miro could be explained by the use of different mutants and raise the possibility that Miro is regulated by accessory proteins bound to the EF hands, or that the regulation of the transport complex is yet to be fully addressed [68]. Indeed, the EF1 domain of Miro1 senses cytosolic calcium and changes the shape of mitochondria independently of MCU-dependent calcium uptake and fusion/fission proteins in fibroblasts [69]. Furthermore, the analysis performed in neurites of iPSCs and in axons and dendrites of primary neurons could also indicate different functions of Miro, depending on its location in neurons. For example, histone deacetylase 6 (HDAC6) deacetylates Miro and stops mitochondria in calcium-rich axons [70] (Figure 2b). Although intriguing, given the broad functions of Mfn2 it is difficult to directly link mitochondrial transport to the levels of mitochondrial calcium [71]. SNPH is another calcium sensor that arrests mitochondria in axons [59,60]. In the presence of elevated calcium, SNPH dissociates Kinesin-1 from mitochondria to enhance their docking at presynapses. High calcium levels disrupt the Miro/Trak/Kinesin complex, favoring the anchoring of mitochondria by SNPH, which inhibits the activity of kinesin [72]. The anchoring of mitochondria to microtubules by SNPH is reversible since in the absence of calcium Miro can rapidly resume its calcium-free conformation and form the Miro/Trak/Kinesin complex to drive mitochondrial motility [62] (Figure 2b).

### 5.2. Glucose

Elevated electric activity mobilizes the Glut3 and Glut4 transporters at presynapses to increase the availability of glucose to boost local metabolism [73,74,75]. Thus, it is not surprising that glucose additionally halts mitochondria via an active mechanism [76]. O-GlcNAcylation is a post-translational modification important for proteins involved in neuronal signaling and synaptic plasticity [77,78]. Glucose activates O-GlcNAc transferase and increases the O-GlcNAcylation of Milton, which inhibits the transport of mitochondria [76]. The four and a half LIM domain protein 2 (FHL2) associates with O-GlcNAcylated Milton and favors the docking of mitochondria to actin [79] (Figure 2c).

### 5.3. ATP

The ratio of ADP and ATP regulates the positioning of mitochondria in primary neurons [63,80], possibly involving mechanisms to sense and drive mitochondria to sites of high energy consumption. Since motor proteins require ATP to transport mitochondria, it is possible that in sites of high ATP consumption, mitochondria halt because kinesin and dynein are not functional. The contributions of ATP and calcium signals to mitochondrial motility are difficult to disentangle because mitochondria are major regulators of the concentration of these two molecules. A study shows that neuronal depolarization decreases ATP levels and activates AMPK, increasing the anterograde transport of mitochondria into axons [80]. Similar results were obtained when lactate uptake was inhibited locally [81]. These studies suggest that the motility of mitochondria can be regulated by ATP independently of calcium. However, it is difficult to understand if this transport is directly regulated by AMPK or if it is the consequence of the broad mechanisms downstream of AMPK (Figure 2c).

### 5.4. Hypoxia

Hypoxia is an important factor during brain ischemia, and it has been shown to regulate mitochondrial motility in neurons [82,83] (Figure 2c). Decreased oxygen induces hypoxia-inducible factor 1α (HIF-1α) which, among its other targets, increases the expression of hypoxia up-regulated mitochondrial movement regulator (HUMMR). Reduced HUMMR in normoxia does not change the motility of mitochondria but diminishes the amount of motile mitochondria with anterograde movement in hypoxia, consistent with a defect in Kinesin-1 transport [82]. Indeed, HUMMR coimmunoprecipitates with Miro and Milton in normoxia [82]. Consistently, the upregulation of HUMMR recovers the average number of axonal mitochondria that are normally reduced by hypoxia [83]. It would be interesting to understand if HUMMR is also a sensor of hypoxia and if it increases its binding to the Miro/Milton complex. Interestingly, nitric oxide (NO) inhibitors recovered the motility of mitochondria during hypoxia and NO administration halted mitochondria [83,84]. It is not clear if this is a direct or indirect effect, nor if HUMMR is involved.

### 5.5. Reactive Oxygen Species (ROS)

Sources of ROS in the CNS are mitochondria and reactive microglia during inflammation. Intracellular and extracellular ROS equally block mitochondria without affecting other organelles [85,86,87]. In these experiments, axons are more vulnerable than dendrites [86]. Two opposing studies show that ROS inhibit the Miro/Milton complex by activating p38α mitogen-activated protein kinase (MAPK) independently from calcium [87] and that ROS increase calcium and the activity of the c-Jun N-terminal Kinase [85] (Figure 2c).

### 5.6. Growth Factors and Neurotransmitters

In addition, molecules for neuronal communication and growth seem to modulate the transport of mitochondria (Figure 2d). In developing neurons, the administration of NGF accumulates mitochondria close to the site of treatment [37]. Although the mechanism is not clear, it seems to involve the phosphoinositide 3-kinase (PI3K) pathway [88]. Moreover, serotonin and dopamine act via their respective receptors on the AKT-GSK3β pathway with opposite effects on mitochondria: dopamine halts and serotonin mobilizes mitochondria [89,90]. Interestingly, GSK3β was found to be localized to HDAC6 and HDAC6 activity and phosphorylation was regulated by GSK3β in primary neurons [91].

## 6. Mitophagy

Mitochondria can also be arrested when they are sequestered and degraded by autophagic engulfment, a process known as mitophagy. Damaged and depolarized mitochondria are cleared by the Pink1/Parkin pathway (Figure 2e). In these conditions, Pink1 is recruited and stabilized by Parkin on the outer mitochondrial membrane. This complex interacts with the Miro/Milton complex and induces the proteasomal degradation of Miro, thus releasing kinesin and arresting mitochondria [92,93]. It is under debate where autophagy occurs, if it involves the transport of mitochondria to the soma or if it happens in distal axons [94,95]. Nevertheless, it is clear that the transport and degradation of mitochondria are intertwined processes regulated by metabolism [96,97].

## 7. Specialized Cytoskeleton Structures Allow Mitochondria to Cross Cell Boundaries

Proper neuronal homeostasis is maintained not only by the intracellular trafficking of mitochondria but also by the intercellular exchange of mitochondria. For example, astrocytes can transfer healthy mitochondria to damaged neurons and provide neuroprotection and neurorepair [7,8,9,98]. Vice versa, astrocytes may internalize damaged mitochondria. Davis and colleagues described this process, named transmitophagy, at the optic nerve head where retinal ganglion cell axons shed the damaged mitochondria to be degraded by astrocytes [99]. Transmitophagy is restricted to a site of high energy demand with limited space, far from the cell body of retinal ganglion cells, suggesting that the cooperation of nearby cells is more advantageous than using resources within the cell to degrade damaged mitochondria. Axonal protrusions in contact with astrocytes contain mitochondria, and microtubules are found proximal to these mitochondria in electron microscopy [99]. Although lacking temporal resolution, this observation leads us to hypothesize the existence of an active mechanism to transfer mitochondria between neural cells. Additional publications show that this mechanism is more widespread in other cells and allow it to be studied in models simpler than the retina. Indeed, stem cells transplanted in vivo have been shown to perform a similar bidirectional mitochondrial exchange [100,101,102]. Therefore, stem cells can both donate their healthy mitochondria and take up damaged mitochondria from stressed somatic cells for degradation [103,104].

### 7.1. Structure of TNTs

While different mechanisms for the transcellular transfer of mitochondria have been identified, including extracellular vesicles and gap junction channels [105], TNTs represent unique actin-rich structures that provide cytoplasmatic continuity between cells, enabling the bidirectional transport of cargoes. TNTs are membrane protrusions linking two or more cells and, as such, they have been described in multiple cell types [106]. They form transient cytoplasmic bridges containing a skeleton mainly composed of F-actin, microtubules or both. Thinner TNTs (<100 nm in diameter) only contain F-actin, whereas thicker TNTs (>100 nm in diameter) are composed of both F-actin and microtubules [107]. Electron and fluorescent microscopy studies performed so far have lacked the spatial resolution to fully investigate the structure and dynamics of TNTs. A recent study using single-molecule-localization-based stochastic optical reconstruction microscopy (STORM) revealed in more detail the organization of TNTs [106]. Interestingly, mitochondria appear to be more abundant in areas closer to cells but also in the bud-shaped sides of TNTs, which are enriched in actin filaments [106]. This observation probably suggests that the movement of mitochondria in TNTs could be regulated in an actin-mediated docking system, as Qin and colleagues speculated in their review [108]. Furthermore, STORM images also reveal that the organization of TNT microtubules differs between cell lines, opening up intriguing scenarios of different mechanisms to traffic mitochondria depending on the type of cell. However, it has to be pointed out that only cell lines have been used in this study and that cells were permeabilized before imaging. Therefore, this method is not suitable to study the motility of mitochondria. So far, it has been shown that TNTs may polymerize and depolymerize rapidly in 30–60 s, spanning distances of up to 300 µm [107], but the use of a higher resolution for a deeper analysis may answer the questions we previously raised.

Prior to the evidence of mitochondrial exchange through membranous channels in 2004 [5,109], Ramírez-Weber and Kornberg provided the first observation of thin actin-based extensions (cytonemes) in disc cells in Drosophila [110]. At that time, the authors suggested that cytonemes might be responsible for some forms of long-range cell–cell communication. To date, microscopy imaging of both live and fixed cells has provided direct evidence of the horizontal transfer of mitochondria through the formation of TNTs under (patho)physiological conditions. Moreover, by combining live imaging and correlative light- and cryo-electron tomography approaches, Sartori-Rupp et al. revealed the ultrastructural features of TNTs. In human and mouse neuronal cells, they proved the different nature of TNTs with respect to other cell protrusions such as filopodia, showing that TNTs form bundles of parallel tubes (iTNTs) braided together by linkers with N-Cadherin, which contain vesicles and mitochondria [111]. While TNTs have been described in many different cell types, no TNT-specific marker has been identified, and they can be recognized based only on morphological characteristics, thus limiting the selective molecular approaches to study TNT formation and the movement of mitochondria [112,113].

### 7.2. Transfer of Mitochondria to Neuronal Cells

It is known that the movement of mitochondria along TNTs is mediated by transport complexes [114]. As for intracellular trafficking, the key protein that modulates intercellular mitochondria transfer is Miro. It interacts directly with the motor protein KIF5, or through adaptor proteins, including Trak1 and Trak2 and Myo10 and Myo19. The knocking down or overexpression of Miro affects the efficiency of the transfer of mitochondria from mesenchymal stem cells (MSCs) to nerve cells in order to repair injury in vitro [102,115]. One fundamental caveat of these experiments is that Miro regulates the mobility of mitochondria within cells and the decreased transfer might be a consequence of a general dysfunction of mitochondrial availability. Furthermore, the downregulation of Miro1 or Miro2 does not completely abrogate the transfer of mitochondria [115]. This observation could be explained by a redundancy of the two proteins, which should be tested by downregulating Miro1 and Miro2 simultaneously in neural cells. Given that Miro mediates the long-range transport of mitochondria using microtubules, it is possible that other mechanisms in the transfer of mitochondria exist, but studies on this matter are still missing. We may also speculate that other components of the well-known system to traffic mitochondria in neurons might be involved and that actin or the orientation of microtubules may provide additional lines of study for mitochondrial transfer between neural cells. Surprisingly, the injection of multipotent mesenchymal stem cells overexpressing Miro1 is able to restore the neurological status of ischemic rats without recovering the ischemic damage [102]. However, this study does not provide evidence of an active transfer of mitochondria in vivo and thus requires further validation. Furthermore, the rescue of neurological functions was measured for 14 days and the efficacy of transplantation in the long term was not evaluated.

### 7.3. The Heterogeneous Nature of TNTs

Although the mechanism regulating the transition of mitochondria entering the TNTs from the cytoplasm is not known, a speculative mechanism has been proposed to be based on an actin docking system [108]. In general, actin has a dominant role in TNTs formation since low doses of the actin-specific inhibitor cytochalasin B are able to dramatically affect the formation of both F-actin and microtubule-based TNTs. Oppositely, microtubule-specific inhibitors do not reduce the formation of TNTs [116,117]. Consistently, regulators of the actin cytoskeleton were identified as key drivers of the membrane nanotube’s formation. For instance, M-Sec (TNFAIP2) in association with Lst1, RelA and other components forms a complex of small GTPases, the exocyst complex, which promotes the formation of TNTs in stressed cells [118,119]. In addition to M-Sec, its interacting protein nucleolin [120], an RNA-binding protein, can promote TNTs formation by binding to and stabilizing the 14–3–3ζ mRNA, thus regulating cortical actin dynamics between primary cortical neurons and astrocytes. Among other proteins known to participate in actin cytoskeleton remodeling, some Rab GTPases have been found to play a role in TNTs formation. Bhat et al. showed that similar to neurite elongation, TNTs formation in neuronal cells is positively regulated by Rab35 through ACAP2 and ARF6-GDP [121]. Ljubojevic et al. recently provided an overview on the actin-related proteins that play a role in TNTs formation [122].

Horizontal mitochondrial transfer has been referred to by Liu et al. as “find me” and “save me” intercellular communication, since damaged cells take up functional mitochondria from healthy donor cells [123]. Cells exposed to ethidium bromide to induce mitochondrial DNA (mtDNA) deletion have been used to prove the intercellular movement of mitochondria from healthy donor cells. Spees et al. demonstrated that these cells that are incapable of aerobic respiration and growth can acquire mtDNA and mitochondria from healthy donor cells and finally regain their oxidative capacity [124]. Consistent with this finding, Lin et al. showed that mtDNA-deficient ρ0 tumor cells co-cultured with mesenchymal cells rescue mitochondrial bioenergetics and oxidative-phosphorylation-dependent cellular growth and motility by acquiring mtDNA [125]. Multiple studies show that the intercellular mitochondrial transfer from healthy donor cells rescues the damage in injured cells [105]. MSCs are the most popular donor cells, indicating that the reparative function in stem cell therapy is partially mediated by mitochondrial transfer [126]. Although in the last decade the number of studies describing mitochondrial transfer has greatly increased, the signaling mechanisms that initiate transcellular mitochondrial trafficking remain largely unknown. For example, it is not clear whether the route is dependent on the recipient damaged or on the healthy donor cell. Several signaling mechanisms of TNTs initiation have been identified. In particular, M-Sec was found to be essential for the formation of TNTs from recipient cells in association with other components forming a complex of small GTPases, the exocyst complex, comprising Cdc42, which is required for the extension of TNTs [118].

In general, the mechanisms of TNTs formation and the initiation factors that drive mitochondrial transfer are still poorly understood and need to be investigated in future studies. Since the intercellular mitochondrial transfer represents what is probably the most exciting therapeutic option for multiple diseases, it will be important to decipher in detail the mechanisms regulating mitochondrial entry into damaged cells, including TNTs formation. Many studies show that it is feasible to treat many diseases in the brain associated with mitochondrial dysfunctions by the described mitochondrial transfer process between cells, or even by the direct transplantation of isolated mitochondria (Table 1).

## 8. Conclusions

In this review, we summarized the complexity of the transport of mitochondria in neurons and the similarity of this system to the intracellular transport of mitochondria. Although the first has been broadly characterized, it could provide a basis for understanding how the latter happens, and ameliorate its potential benefits in therapeutic approaches. The current understanding of this mode of transport is quite extensive, but it is clear that some of the conclusions established in the past were oversimplified. Recent advances in technology and analysis will help to investigate the fine regulation of mitochondrial motility, especially regarding poorly studied movements such as the duration of pauses and oscillatory movements. These mechanisms will possibly link microtubule dynamics to motor proteins at a more refined resolution, scaling down to micron-size compartments such as spines. It will be intriguing to understand how metabolism regulates such movement at a single-mitochondrion scale.

## Figures and Tables

**Figure 1 biomolecules-13-00938-f001:**
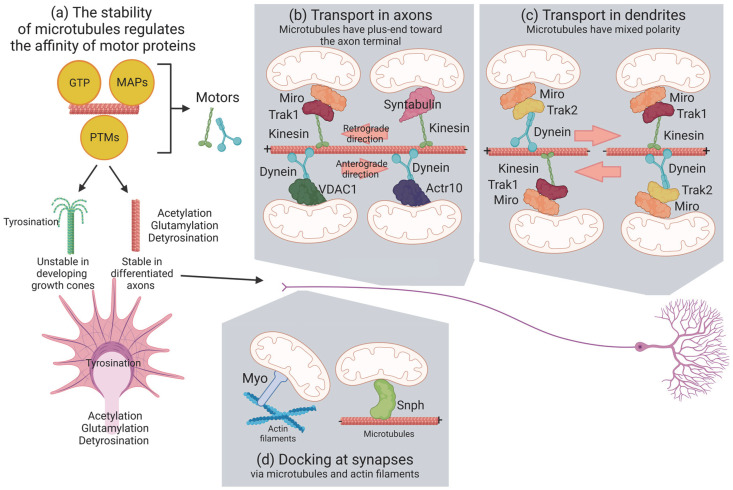
Organization of cytoskeleton and molecular components for the transport of mitochondria in neurons. Neurons are polarized cells divided into soma, dendrites and axons. The long-range transport of organelles is mediated via motor proteins, which bind microtubules. The regulation of this transport is regulated by the stability of microtubules (**a**) and by specific adaptors (**b**–**d**). The stability of microtubules increases the affinity of motors and enhances the transport of organelles and cargoes. Guanosine triphosphate (GTP), post-translational modifications (PTMs) and microtubule-associated proteins (MAPs) are the main factors that regulate such stability, which is key during the development of axons (**a**). For instance, tyrosinated microtubules are unstable at the distal tips of axons, named growth cones, which respond to guidance clues to remodel the actin and microtubule cytoskeleton to orient the growth of axons. Once the axon is extended, microtubules are acetylated, glutamylated and detyrosinated to promote stability and a fixed orientation in mature neurons. Indeed, microtubules are plus-ended, oriented toward axonal terminals and are of mixed orientation in dendrites (**b**,**c**). The orientation of microtubules in axons determines the transport of mitochondria towards the distal part (retrograde) or the soma (anterograde) (**b**). The motor proteins kinesin and dynein mediate the anterograde and retrograde transport of organelles and vesicle cargoes, respectively. Specific receptors link these motor proteins to mitochondria. In axons, kinesin binds to the Miro/Trak1 complex or to Syntabulin to transport mitochondria towards the axon terminal, while actin-related protein 10 (Acrt10) and voltage-dependent anion-selective channel (VDAC1) link mitochondria to the dynein/dynactin complex in the anterograde direction (**b**). The Miro/Traks complex regulates the transport of mitochondria in dendrites. Trak1 and Trak2 mediate the retrograde and anterograde directions, respectively (**c**). Mitochondria also travel along actin filaments, but the mechanisms regulating long-range transport are largely unknown. However, actin filaments regulate a third mechanism to arrest and dock mitochondria at sites with high energy demands, such as synapses, using the adaptors Myo6 or 19 and Syntaphilin (Snph) (**d**). Created with Biorender.com.

**Figure 2 biomolecules-13-00938-f002:**
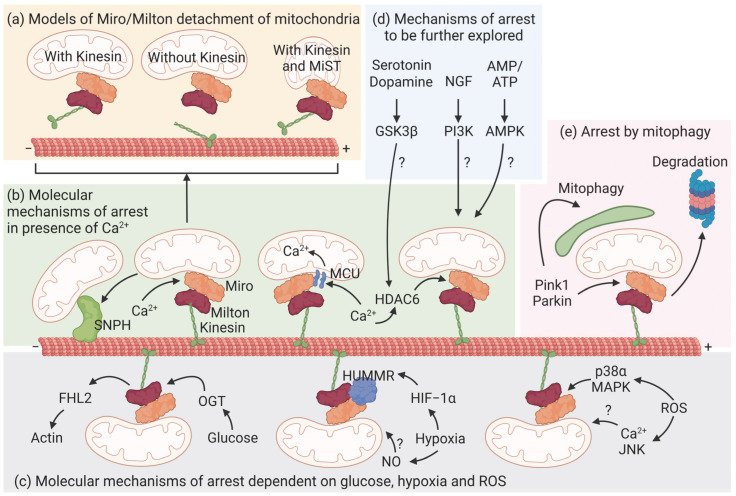
Regulation of mitochondrial transport. Mitochondria are transported along microtubules with the kinesin motor and the Milton/Miro adaptors organized in a complex. Mitochondria detach from microtubules when this complex is not active. The detailed mechanism of detachment is not yet clear. Indeed, it has been proposed that mitochondria halt following the dissociation of the whole complex, or that kinesin remains attached to microtubules, possibly involving a change in the shape of mitochondria (mitochondrial shape transition, MiST) independent of fusion and fission proteins (**a**). Nevertheless, multiple studies show that the main regulator of the activity of the Kinesin/Milton/Miro complex is calcium (Ca^2+^). Additionally, in this case, studies are contradictory. It is reported that cytoplasmatic calcium binds directly to Miro EF hands and induces a conformational change in Miro, thus arresting mitochondria (**b**). Alternatively, mitochondria can also arrest when the amount of calcium increases in the matrix through the mitochondrial calcium uniporter (MCU). In addition, mitochondria can stop following the deacetylation of Miro by histone deacetylate 6 (HDAC6) in the presence of cytosolic calcium (**b**). Moreover, mitochondria can be transferred to the adaptor Syntaphilin (SNPH) after dissociation from the Kinesin/Milton/Miro complex to enhance their docking in synapses (b). Other metabolic pathways can arrest mitochondria in neurons (**c**): glucose induces the O-GlcNAcylation of Miro and the recruitment of four and a half LIM domain protein 2 (FHL2) to dock mitochondria to actin filaments. Hypoxia increases the levels of nitric oxide (NO) and induces the expression of hypoxia up-regulated mitochondrial movement regulator (HUMMR), which interacts with the Milton/Miro complex to regulate kinesin-mediated transport. Furthermore, reactive oxygen species (ROS) activate p38α mitogen-activated protein kinase (MAPK), which inhibits Miro and activates c-Jun N-terminal Kinase (JNK) to stop mitochondria. There are other mechanisms that regulate the motility of mitochondria that remain to be investigated in detail (**d**). For instance, the neurotransmitters serotonin and dopamine are opposite modulators of mitochondrial motility, possibly via glycogen synthase kinase 3β (GSK3β). Moreover, nerve growth factor (NGF) via phosphatidylinositol-3-kinase (PI3K) and the ratio of AMP/ATP levels sensed by AMP-activated protein kinase (AMPK) regulate the movement of mitochondria by unknown mechanisms. Finally, depolarized and damaged mitochondria can recruit and stabilize the Pink1/Parkin complex, which targets Miro for proteasomal degradation (**e**). Thus, mitochondria stop and are degraded by autophagy. Created with Biorender.com.

**Table 1 biomolecules-13-00938-t001:** Mitochondrial transfer therapy for brain diseases.

Disease	Treatment	Clinical Outcome	Ref.
Rat model of Parkinson’s disease	Mitochondria	Restored mitochondrial functions and reduced oxidative damage in dopaminergic neurons	[127]
Mouse model of PD	Mitochondria	Increased electron transport chain activity, reduced ROS level and prevented apoptosis and necrosis	[128]
Rat model of schizophrenia	Mitochondria	Prevented mitochondrial dysfunction in intra-prefrontal cortex neurons and emergence of attention deficit	[129]
Middle cerebral artery occlusion (MCAO) in rats	Mitochondria	Decreased brain infarct volume and reversed neurological deficits.	[130]
MCAO in rats	Mesenchymal multipotent stromal cells	Reduced infarct volume in the brain and partial restoration of neurological status *	[131]
Ischemic stress in rats	Mitochondria	Restored motor performance, attenuated brain infarct area and neuronal cell death	[132]
MCAO in rats	MSC-derived mitochondria	Declined blood creatine phosphokinase level, abolished apoptosis, decreased astroglyosis and microglia activation, reduced infarct size and improved motor function	[133]
Ischemia–reperfusion stroke injury	MSCs	Rescued damaged cerebrovascular system in stroke	[123]
Spinal cord injury in rats	Mitochondria	Maintenance of normal bioenergetics without recovery of motor and sensory functions	[134]
Traumatic brain injury in rats	MSC-derived mitochondria	Improved sensorimotor functions	[135]
Nerve crush injury in rats	Mitochondria	Improved neurobehaviors, electrophysiology of nerve conduction and muscle activities	[136]

* More profound neuroprotective effects have been obtained when MSCs were injected after cocultivation with neurons.

## Data Availability

No new data were created or analyzed in this study. Data sharing is not applicable to this article.

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
