# Peer review of "Highly Specialized Mechanisms for Mitochondrial Transport in Neurons: From Intracellular Mobility to Intercellular Transfer of Mitochondria"

_biomolecules, 2023, doi:10.3390/biom13060938_

Round 1

Reviewer 1 Report

The manuscript presented by Zaninello and Bean provides a comprehensive and well-organized review on mitochondrial transport, with a specific focus on neurons. Although it is well presented and organized, and it reaches interesting conclusions, there are some aspects that need to be fixed before its publication.
The figures lack detail in the described processes, as well as in the figure captions. The figures, along with their captions, should be self-explanatory.
It would be necessary to include a figure that depicts the growth cone and neuron differentiation, or alternatively, incorporate it into one of the existing figures. Additionally, more references to the figures need to be made in the text when the porcesses are described.
Table 1 would be better placed at the beginning of page 9.

Author Response

We are delighted to read that the Reviewer has found our manuscript “well-presented and organized”, and that it has been considered to provide a comprehensive review on mitochondrial transport with a specific focus on neurons. We agree with the Reviewer that “the figures, along with their captions should be self-explanatory”. For this reason, we modified the figures by including subtitles and letters in order to increase their readability and make more clear the described processes. Accordingly, we also modified captions and we added in the main text more references to the figures. We believe that the figures are now ameliorated and better referred in the main text of the revised version of our manuscript.

We thanks the Reviewer who suggested to “incorporate the growth cone and axonal differentiation in one of the existing figures”. We did it within the Figure 1.

Finally, we moved Table1 as at the beginning of page 9 as per this Reviewer suggestion.

Reviewer 2 Report

This is a compact review of mitochondrial motility in neurons and how it is controlled. Part I deals with intraneuronal movement, Part II with transcellular mitochondrial exchange. While Part I is extensive and comprehensive, Part II is a little short. In addition, we suggest that the presentation of Part II should include relevant methods by which mitochondrial exchange has been verified, or at least a critical view presenting the evidence.

Minor:
1. page 3 line 135: it has to be KDa.
Line 144: it has to be 1.2 MDa
2. please add a short description to figure 1
3. page 7, line 301: it has to be "mitochondria"
4. Hypoxia: please explicitly explain what upregulation (in addition to downregulation) of HUMMR does (section 5.4)
5. Please provide a list of abbreviations
6. move the text below figure 2 to the legend of figure 2, this would help understanding

English is fine

Author Response

This is a compact review of mitochondrial motility in neurons and how it is controlled. Part I deals with intraneuronal movement, Part II with transcellular mitochondrial exchange. While Part I is extensive and comprehensive, Part II is a little short. In addition, we suggest that the presentation of Part II should include relevant methods by which mitochondrial exchange has been verified, or at least a critical view presenting the evidence.

We thank the Reviewer for his/her comments and suggestions that allow us to improve our manuscript. As suggested by the Reviewer, we extended the Part II about the transcellular mitochondrial exchange. Moreover, we included in the revised version of our manuscript the relevant methods by which the mitochondrial exchange has been verified.

Minor:
1. page 3 line 135: it has to be KDa.
Line 144: it has to be 1.2 MDa

Thanks. We corrected these mistakes.

  1. please add a short description to figure 1

Done.

  1. page 7, line 301: it has to be "mitochondria"

Thanks. We corrected this mistake.

  1. Hypoxia: please explicitly explain what upregulation (in addition to downregulation) of HUMMR does (section 5.4)

We thank the Reviewer for the comment. We integrated the new information in the text.

  1. Please provide a list of abbreviations

Done.

  1. move the text below figure 2 to the legend of figure 2, this would help understanding

Done.